# EXPLICIT HOMOGRAPHY ESTIMATION IMPROVES CONTRASTIVE SELF-SUPERVISED LEARNING

## ABSTRACT

The typical contrastive self-supervised algorithm uses a similarity measure in latent space as the supervision signal by contrasting positive and negative images directly or indirectly. Although the utility of self-supervised algorithms has improved recently, there are still bottlenecks hindering their widespread use, such as the compute needed. In this paper, we propose a module that serves as an additional objective in the self-supervised contrastive learning paradigm. We show how the inclusion of this module to regress the parameters of an affine transformation or homography, in addition to the original contrastive objective, improves both performance and learning speed. Importantly, we ensure that this module does not enforce invariance to the various components of the affine transform, as this is not always ideal. We demonstrate the effectiveness of the additional objective on two recent, popular self-supervised algorithms. We perform an extensive experimental analysis of the proposed method and show an improvement in performance for all considered datasets. Further, we find that although both the general homography and affine transformation are sufficient to improve performance and convergence, the affine transformation performs better in all cases.

## 1 INTRODUCTION

There is an ever-increasing pool of data, particularly unstructured data such as images, text, video, and audio. The vast majority of this data is unlabelled. The process of labelling is time-consuming, labour-intensive, and expensive. Such an environment makes algorithms that can leverage fully unlabelled data particularly useful and important. Such algorithms fall within the realm of unsupervised learning. A particular subset of unsupervised learning is known as Self-Supervised Learning (SSL). SSL is a paradigm in which the data itself provides a supervision signal to the algorithm.

Somewhat related is another core area of research known as transfer learning (Wang et al., 2020). In the context of computer vision, this means being able to pre-train an encoder network offline on a large, varietal dataset, followed by domain-specific fine-tuning on the bespoke task at hand. The state-of-the-art for many transfer learning applications remains dominated by supervised learning techniques (Tan et al., 2020; Martinez et al., 2019; Donahue et al., 2014; Girshick et al., 2014), in which models are pre-trained on a large labelled dataset.

However, self-supervised learning techniques have more recently come to the fore as potential alternatives that perform similarly on downstream tasks, while requiring no labelled data. Most self-supervised techniques create a supervision signal from the data itself in one of two ways. The one approach are techniques that define a pre-text task beforehand that a neural network is trained to solve, such as inpainting (Pathak et al., 2016) or a jigsaw puzzle (Noroozi & Favaro, 2016). In this way, the pre-text task is a kind of proxy that, if solved, should produce reasonable representations for downstream visual tasks such as image or video recognition, object detection, or semantic segmentation. The other approach is a class of techniques known as contrastive methods (Chen et al., 2020a; He et al., 2019; Chen et al., 2020b). These methods minimise the distance (or maximise the similarity) between the latent representations of two augmented views of the same input image, while simultaneously maximising the distance between negative pairs. In this way, these methods enforce consistency regularisation (Sohn et al., 2020), a well-known approach to semi-supervised learning. These contrastive methods often outperform the pre-text task methods and are the current state-of-the-art in self-supervised learning. However, most of these contrastive methods have several

drawbacks, such as requiring prohibitively large batch sizes or memory banks, in order to retrieve the negative pairs of samples (Chen et al., 2020a; He et al., 2019).

The intuition behind our proposed module is that any system tasked with understanding images can benefit from understanding the geometry of the image and the objects within it. An affine transformation is a geometric transformation that preserves parallelism of lines. It can be composed of any sequence of rotation, translation, shearing, and scaling. A homography is a generalisation of this notion to include perspective warping. A homography need not preserve parallelism of lines, however, it ensures lines remain straight. Mathematically, a homography is shown in Equation 1. It has 8 degrees of freedom and is applied to a vector in homogenous coordinates. An affine transformation has the same form, but with the added constraint that $\phi_{3,1} = \phi_{3,2} = 0$.

$$H_\phi = \begin{bmatrix} \phi_{1,1} & \phi_{1,2} & \phi_{1,3} \\ \phi_{2,1} & \phi_{2,2} & \phi_{2,3} \\ \phi_{3,1} & \phi_{3,2} & 1 \end{bmatrix} \tag{1}$$

The ability to know how a source image was transformed to get to a target image implicitly means that you have learned something about the geometry of that image. An affine transformation or, more generally, a homography is a natural way to encode this idea. Forcing the network to estimate the parameters of a random homography applied to the source images thereby forces it to learn semantics about the geometry. This geometric information can supplement the signal provided by a contrastive loss, or loss in the latent space.

In this paper, we propose an additional module that can be used in tandem with contrastive self-supervised learning techniques to augment the contrastive objective (the additional module is highlighted in Figure 1). The module is simple, model-agnostic, and can be used to supplement a contrastive algorithm to improve performance and supplement the information learned by the network to converge faster. The module is essentially an additional stream of the network with the objective of regressing the parameters of an affine transformation or homography. In this way, there is a multi-task objective that the network must solve: 1. minimising the original contrastive objective, and 2. learning the parameters of a homography applied to one of the input images from a *vector difference* of their latent representations. We force the latent space to encode the geometric transformation information by learning to regress the parameters of the transformation in an MLP that takes the vector difference of two latent representations of an input, $x$, and its transformed analogue, $x'$. By including the information in this way, the network is *not* invariant to the components of the transformation but is still able to use them as a self-supervised signal for learning. Moreover, this approach serves as a novel hybrid of the pre-text tasks and contrastive learning by enforcing consistency regularisation (Sohn et al., 2020).

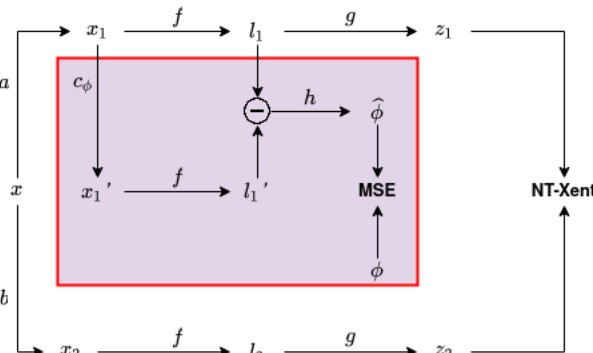

Figure 1: Proposed architecture. The highlighted box highlights the additional proposed module tasked with regressing the parameters of an affine transformation or homography.

Through extensive empirical studies, we show that the additional objective of regressing the transformation parameters serves as a useful supplementary task for self-supervised contrastive learning, and improves performance for all considered datasets in terms of linear evaluation accuracy and convergence speed.

The remainder of the paper is structured as follows. In Section 2, we cover the related work in the area of self-supervised learning, going into detail where necessary. In Section 3 we detail our

proposed method. We first introduce a framework and set of notation to make the formalisation of the approach clear. We then delve into the details behind the architecture and choices for the various part of the system. This is followed by a comprehensive set of experiments in Section 4, including results of various datasets, as well as an ablative study. Finally, the paper is concluded with some closing remarks in Section 5.

## 2 RELATED WORK

SSL is a popular research area within computer vision. Previous approaches can be broadly classed into two main categories. The first is where pre-text tasks are manually defined, and the goal of the algorithms is to solve these hand-crafted tasks (Lee et al., 2020; Doersch et al., 2015; Gidaris et al., 2018; Zhang et al., 2016; Misra & Maaten, 2020). Examples of such methods include inpainting (Pathak et al., 2016), colourising (Zhang et al., 2016), jigsaw puzzles (Noroozi & Favaro, 2016), patch prediction (Doersch et al., 2015), and geometric image transformations (Dosovitskiy et al., 2014) such as using rotation as the pre-text task (Gidaris et al., 2018; Feng et al., 2019). Some of these pre-text approaches that deal with geometric image transformations are similar in spirit to our method. Gidaris et al. (2018); Feng et al. (2019) are two variants of predicting image rotations as an auxiliary task for unsupervised learning. Perhaps closer to our method is Dosovitskiy et al. (2014), in which a set of transformations is applied to image patches, and the network is trained in a fully-unsupervised manner to predict surrogate classes defined by a set of transformed image patches by minimising the log loss. Our method, however, investigates a different, particular set of transformations (those that define an affine transformation of general homography), and show this can be used to aid self-supervised performance, using the transformation parameters themselves as targets that need to be regressed (using mean-squared error) by the contrastive algorithm in a multi-task manner. The discrepancy in the network's ability to predict the actual values of the parameters of the affine transformation/homography serves as our additional supervision signal.

A somewhat related approach to our proposed method within the pre-text task domain is proposed by Lee et al. (2020). They propose to augment the learning process of a supervised learning algorithm with additional labels constructed using self-supervised labels. These labels are rotation classes and colour permutations. Importantly, they create a loss function which is based on a joint distribution of the original (supervised) labels and the self-supervised (augmented) labels. In this way, the network is not forced to be invariant to the transformations under consideration, since this has been shown to hurt performance (Lee et al., 2020). Our method is different to this in that we propose a module to be integrated specifically with self-supervised algorithms. Additionally, we regress the transformation parameters in real vector space and do not create classes for the parameters.

The other broad category of SSL is based on contrastive learning (Chen et al., 2020a; He et al., 2019; Caron et al., 2020), and this class of techniques represent the current state-of-the-art in self-supervised learning, outperforming the hand-crafted pre-text task methods. These approaches learn representations by contrasting positive pairs of samples from negative pairs of samples in latent space. Such methods typically require that careful attention be paid to the negative samples. Additionally, they have the disadvantage of requiring prohibitively large batch sizes (4096-16000), memory banks, or other mechanisms to retrieve the relevant negative samples.

One popular such method is known as SimCLR (Chen et al., 2020a). SimCLR is a general framework for contrastive learning, and in its vanilla formulation consists of an encoder network parameterised by a CNN (usually a variant of ResNet (He et al., 2016)) and an MLP projection head. An input image is sampled, and two distinct views of that same input image are computed using a random augmentation. The augmentation consists of colour jiterring, Gaussian blurring, and random cropping. The two views are sent through the encoder network to produce two latent representations. These latent vectors are then sent through the projection head to produce final latent vectors. It is from these vectors that the loss is computed. In the case of SimCLR, the loss is normalised temperatured cross-entropy (NT-Xent).

A recent approach proposed in Grill et al. (2020) (BYOL) somewhat overcomes the aforementioned disadvantages of requiring negative pairs of samples (which implicitly requires a large batch size). Two separate networks with their own weights are used in tandem to learn the representation. An *online* network (consisting of an encoder, MLP projection head, and MLP prediction network) is trained to predict the representation outputted by a *target* network. During training, the online

network parameters are updated using backpropagation of error derivatives computed using a mean-squared error loss. However, the target network parameters are updated using an exponential moving average. In this way, BYOL overcomes collapsed solutions in which every image produces the same representation. We test our module with both SimCLR and BYOL, since these two methods serve as two popular, recent approaches to contrastive SSL.

Some helpful findings for guiding self-supervised research were demonstrated in Kolesnikov et al. (2019). Core among these are that 1) standard architecture designs that work well in the fully-supervised setting do not necessarily work well in the self-supervised setting, 2) in the self-supervised setting larger CNNs often means higher quality learned representations, and 3) the linear evaluation paradigm for assessing performance may take a long time to converge. Moreover, Newell & Deng (2020) find that the effectiveness of self-supervised pretraining decreases as the amount of labelled data increases, and that performance on one particular downstream task is not necessarily indicative of performance on other downstream tasks.

## 3 PROPOSED METHOD

We first introduce a mathematical framework for discussing our method. Let $\mathcal{B}_1$ be a set of *base transformations*. A base transformation is a transformation that cannot be decomposed into more basic transformations and is interpreted as per Grill et al. (2020); Chen et al. (2020a). Examples of base transformations include colour jittering, cropping, and horizontal flipping. We define the possible base transformations a-priori, and $|\mathcal{B}_1| < \infty$. Next, we define a new set of base spatial transformations $\mathcal{B}_2$ that correspond to the general affine transformations (i.e. rotation, translation, scaling and shearing) or the full homography (i.e. affine transformations and perspective projection). Further, we impose the following condition:

$$\mathcal{B}_1 \cap \mathcal{B}_2 = \emptyset \tag{2}$$

The reason for this restriction will be apparent later.

A transformation $t_{b,\theta}$ is parameterised by its associated base transformation $b \in \mathcal{B}_1 \cup \mathcal{B}_2$ and transformation parameters $\theta \in \Theta$. Then, the set of all possible transformations for a particular base transformation set $\mathcal{B}$ may be defined as:

$$\mathcal{T}_i := \{t_{b,\theta} | b \in \mathcal{B}_i, \theta \in \Theta\} \tag{3}$$

Clearly, we may have that $|\mathcal{T}_i| = \infty$, since some parameters may take on any value within compact subsets of $\mathbb{R}$. This is important because we want to be able to sample from an infinite sample space during training to ensure the network sees a variety of samples.

We can now define an *augmentation*, which is an *ordered* sequence of $n$ transformations. As such, each unique ordering will necessarily produce a unique augmentation (e.g. flipping and then cropping is different from cropping and then flipping). Formally, an augmentation $a$ is defined as:

$$a(x) = t_{b_n,\theta_n} \circ \cdots \circ t_{b_2,\theta_2} \circ t_{b_1,\theta_1}(x) \tag{4}$$

Denote the set of all possible augmentations for a transformation set $\mathcal{T}_i$ as $\mathcal{A}_{\mathcal{T}_i}$. Under this definition, $\mathcal{A}_{\mathcal{T}_2}$ is the set of all possible affine or homographic transformations. Examples of the affine transformations and homographies can be seen in Appendix A.

Now, consider an input image $x$ sampled at random from a dataset of images $X \subset \mathcal{X}$, where $\mathcal{X}$ is the sample space of images. We sample augmentations $a, b \in \mathcal{A}_{\mathcal{T}_1}$, and apply them to $x$ to produce augmented views $x_1$ and $x_2$, respectively. We then sample an affine/homographic transformation $c_\phi \in \mathcal{A}_{\mathcal{T}_2}$ and apply it to $x_1$ to produce $x_1'$. Note that $x_1$ and $x_1'$ are related by a homography. This is a core assumption relied upon by further inductive biases we introduce into our model.

We now describe the proposed architecture as depicted in Figure 1. Let the mapping $f : \mathcal{X} \to \mathbb{R}^p$ be parameterised by a CNN, and the mappings $g : \mathbb{R}^p \to \mathbb{R}^k$ and $h : \mathbb{R}^p \to \mathbb{R}^m$ be parameterised by MLPs, where $p$, $k$, and $m$ are the dimensionality of the encoder latent vector, projection head latent vector, and homography parameter vector, respectively. $f$ and $g$ are the encoder and projection head from the original SimCLR (Chen et al., 2020a) and BYOL (Grill et al., 2020) formulations, respectively, whereas $h$ is a new MLP tasked with estimating the homography parameters. Note that

if we are regressing all parameters of a general affine transformation, then $m = 6$, whereas for a full homography we have $m = 8$. For brevity, we have denoted both streams in the architecture to be a network with the same shared weights, although it may be the case that the two streams consist of networks with different weights (as is the case with BYOL).

The loss function for our method contains two terms. First is the original loss function as defined by the original method: NT-Xent for SimCLR and squared $L_2$ for BYOL. We define this first term as $\mathcal{L}_1(z_1, z_2)$, where $z_1 = g(f(x_1))$ and $z_2 = g(f(x_2))$. The second term can be seen as forcing the network to explicitly learn the affine transformation or homography between $x_1$ and $x'_1$. Let the latent representations of $x_1$ and $x'_1$ be $l_1 = f(x_1)$ and $l'_1 = f(x'_1)$. We send the vector difference $l_1 - l'_1$ through $h$ to produce an estimate of the homography's parameters. We regress to these parameters using mean-squared error: $\mathcal{L}_2(h(l_1 - l'_1), \phi)$, where $\phi$ are the ground truth affine transformation parameters. Thus, the complete loss function is given by:

$$\mathcal{L}_1(z_1, z_2) + \mathcal{L}_2(h(l_1 - l'_1), \phi) \tag{5}$$

The vector difference naturally describes the transformation needed to move from $l_1$ to $l'_1$. With our architecture and learning objective, we force this vector difference transformation vector to encode the homography between $x_1$ and $x'_1$. This interpretation may be seen as natural and intuitive. Hence, the $\mathcal{L}_1$ term enforces invariance to the transformations in $\mathcal{B}_1$ and $\mathcal{L}_2$ enforces non-invariance to the transformations in $\mathcal{B}_2$.

Note that this is still completely self-supervised. Moreover, the restriction imposed in Equation 2 is necessary because we cannot have any transformations in $c_\phi$'s sequence that would destroy the fact that $x_1$ and $x'_1$ are related through a homography. For example, adding a *cropping* transformation would break the homography assumption. One may add transformations that do not break this restriction (e.g. colour jitter), however, we do not explore this here.

We may interpret this extended architecture as solving a multi-task learning objective in which 1) the contrastive loss between differing augmented views of the image must be minimised, and 2) another network must be able to estimate the homography between images, which explicitly forces the latent space to encode this spatial information during training.

## 4 EXPERIMENTS

### 4.1 EXPERIMENTAL SETUP

This section presents an empirical study comparing the original SimCLR and BYOL techniques on the CIFAR10, CIFAR100 and SVHN benchmark datasets, with and without our proposed module. Our goal is *not* to achieve near state-of-the-art performance on the datasets, but is rather to demonstrate the effectiveness of the proposed additional homography estimation objective under consistent experimental settings. In all cases, the proposed module improves the performance of a linear classifier on the learned representation and improves the learning speed.

The experimental setup for the self-supervised training of SimCLR and BYOL can be found in Table 1. The batch size is somewhat lower than the original methods since the original methods focused on performance on ImageNet, which requires a considerably larger batch size to perform well. In some additional experiments, we find performance decreased for our datasets with batch sizes larger than 256 for all methods (original SimCLR and BYOL, as well as our method). Further, we found alternative optimised hyperparameter values (learning rate, optimiser, and weight decay) that worked better than those proposed in the original formulations of SimCLR and BYOL, which can be attributed to similar reasons as the batch size arguments. We use the same type of learning rate decay as the previous methods, and train for the same number of epochs (and warmup epochs) as SimCLR. We use a temperature of 0.5 for the NT-Xent loss and keep all images at their default resolution of $32 \times 32$. Lastly, all reported confidence intervals are the average across 10 trials of the full pipeline trained from scratch (SSL pretraining + linear evaluation).

Performance is measured as per the literature, using linear evaluation on the relevant dataset. The experimental setup for linear evaluation can be seen in Table 1. We freeze the encoder and only optimise the weights of a final linear layer using cross-entropy.

Table 1: Experimental setup.

| SSL | | | | Linear Evaluation | | |
|---|---|---|---|---|---|---|
| | SimCLR | BYOL | | | SimCLR | BYOL |
| Batch size | 256 | 256 | | Batch size | 64 | 64 |
| Optimiser | Adam | SGD | | Optimiser | Adam | Adam |
| LR | 3e-04 | 0.03 | | LR | 3e-04 | 3e-04 |
| Momentum | - | 0.9 | | Epochs | 200 | 200 |
| Weight decay | 10e-06 | 4e-04 | | Hardware | | |
| Epochs (warmup) | 100 (10) | 100 (10) | | V100 16GB GPU | | |
| LR schedule | Cosine decay | Cosine decay | | | | |

We parameterise $f$ as ResNet50, while $g$ and $h$ are parameterised as two-layer ReLU MLPs (Figure 1). Further, to ensure consistency with SimCLR, we have that $\mathcal{B}_1 = \{\text{random crop}, \text{random horizontal flip}, \text{colour jitter}, \text{Gaussian blur}, \text{random grayscale}\}$. The output of $h$ is a six-dimensional real vector, where the six components are defined according to the parameters of a general affine transform: 1) rotation angle, 2) vertical translation, 3) horizontal translation, 4) scaling factor, 5) vertical shear angle, and 6) horizontal shear angle. For a homography, the output of $h$ is instead an eight-dimensional vector. For details about the transformations, see Appendix A.

## 4.2 AFFINE AND HOMOGRAPHY OBJECTIVE

From Tables 2 and 3 ('+ H' and '+ A' for homography and affine, respectively) we can see that the estimation of the affine transformation and the homography both assist performance and allow for faster learning. In particular, we note statistically significant improvements across all datasets for both SimCLR and BYOL with the affine objective.[1] We posit that the ability to explicitly estimate the affine transformation or homography between input images in this way allows the encoder to learn complementary information early on in training that is not available from the contrastive supervision signal. The ability to estimate the affine transform or homography means that the network is encoding the geometry of the input images. This explicit geometric information is not directly available from the contrastive signal. Interestingly, the affine objective outperforms the full homography in all cases, even though an affine transformation is a subset of a homography. We perform a sweep of the distortion amount for the homography and find it consistently performs similar to or a little worse than the affine transform (see Appendix B). When the distortion factor becomes too large, accuracy drops noticeably as the images are too distorted to learn effectively. We note that incorporating our module into a network results in an average 30% additional training time versus the respective original methods.

Table 2: Performance with SimCLR on various datasets (mean $\pm$ 99% confidence interval).

| | CIFAR10 | CIFAR100 | SVHN |
|---|---|---|---|
| SimCLR | $63.34 \pm 0.0016$ | $28.53 \pm 0.0017$ | $83.19 \pm 0.0017$ |
| SimCLR + H | $64.04 \pm 0.0029$ | $29.10 \pm 0.0025$ | $82.37 \pm 0.0024$ |
| SimCLR + A | $\mathbf{64.71 \pm 0.0023}$ | $\mathbf{31.33 \pm 0.0024}$ | $\mathbf{83.85 \pm 0.0017}$ |

Table 3: Performance with BYOL on various datasets (mean $\pm$ 99% confidence interval).

| | CIFAR10 | CIFAR100 | SVHN |
|---|---|---|---|
| BYOL | $56.56 \pm 0.0026$ | $23.25 \pm 0.0026$ | $72.34 \pm 0.0047$ |
| BYOL + H | $58.78 \pm 0.0033$ | $25.88 \pm 0.0029$ | $76.56 \pm 0.0054$ |
| BYOL + A | $\mathbf{60.19 \pm 0.0016}$ | $\mathbf{28.10 \pm 0.0018}$ | $\mathbf{78.71 \pm 0.0050}$ |

Figure 2 shows the linear evaluation accuracy trained on the embeddings extracted from the model at each epoch during the SSL training. We can see that performance and convergence improves with the inclusion of the proposed module. The module and its accompanying additional objective of

---

[1]Using a t-test and significance level of 1%

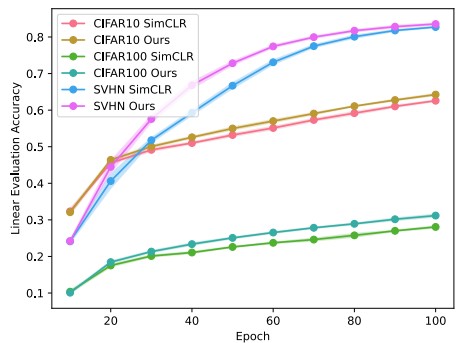 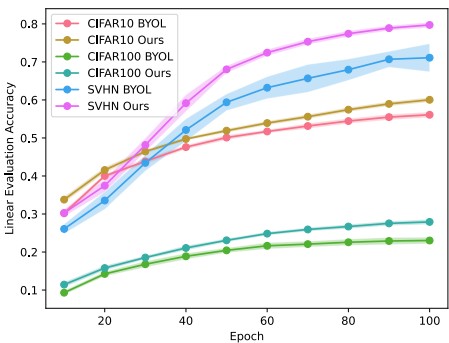

(a) SimCLR results (measured every 10 epochs).      (b) BYOL results (measured every 10 epochs).

Figure 2: Graph of the linear evaluation accuracy at various points during training for both Sim-CLR and BYOL with and without our proposed module. The shaded region indicates one standard deviation from the mean.

regressing the affine transform/homography may be seen as a regulariser for the original contrastive objective. This is further evidenced by the shaded regions in the figures, in which the proposed method results in more stable performance.

We note that the relative benefit of our proposed module diminishes with longer training time for SimCLR and BYOL. This makes sense as the relative benefit of the module decreases with time as the model learns to estimate the affine transformation or homography more accurately as the epochs progress. We performed additional experiments on SimCLR and BYOL training the model for longer, and note that the proposed module still outperforms or performs similarly to the original methods on all datasets. This is shown in Table 4. These results also verify the findings of previous works that find that larger models trained for longer benefits self-supervised architectures (Chen et al., 2020a; Grill et al., 2020; Chen et al., 2020b; Kolesnikov et al., 2019).

Table 4: Performance comparison of SimCLR and BYOL on various datasets trained for 500 epochs.

|  | CIFAR10 | CIFAR100 | SVHN |
|---|---|---|---|
| SimCLR | $76.96 \pm 0.0031$ | $42.17 \pm 0.0001$ | $88.37 \pm 0.0030$ |
| SimCLR + A | $76.80 \pm 0.0020$ | $\mathbf{42.53 \pm 0.0014}$ | $88.83 \pm 0.0015$ |
| BYOL | $\mathbf{78.00 \pm 0.0060}$ | $38.05 \pm 0.0007$ | $\mathbf{90.78 \pm 0.0067}$ |
| BYOL + A | $77.45 \pm 0.0020$ | $40.91 \pm 0.0005$ | $90.74 \pm 0.0025$ |

We note that the performance gap between SimCLR and BYOL evident in Tables 2 and 3 in general can be attributed to the fact that in the original works, BYOL was trained for 10x as long as SimCLR, whereas we trained both for the same number of epochs as the original SimCLR work. We posit that BYOL has simply not converged sufficiently, since BYOL eventually outperforms SimCLR (as evidenced by Table 4). This is consistent with the findings from the original works.

### 4.3 INVARIANCE IS NOT ALWAYS DESIRABLE

In order for a function $f$ to be invariant to a transformation $T$, we must have that, for all $x$, $f(x) = f(Tx)$. Thus, one way to encourage invariance to $T$ in a neural network $f$ is to add a term to the loss function which minimises:

$$L(f(x), f(Tx)) \tag{6}$$

for some measure of similarity $L$. If we rewrite our loss function from Equation 5 in terms of our input image $x$ and augmentations $a, b, c_\phi$, we get:

$$\mathcal{L}_1(g(f(ax)), g(f(bx))) + \mathcal{L}_2(h(f(ax) - f(c_\phi ax)), \phi) \tag{7}$$

The first term of the above loss, corresponding to the SimCLR/BYOL loss, is clearly of the form of Expression 6. This means that we are encouraging our representation to be invariant to the transformations within $\mathcal{B}_1$. However, the second term in the loss (i.e. the term corresponding to the affine

transformation/homography parameter estimation) is not of the form of Expression 6, since we have recast the objective into a parameter prediction task. Thus, we are not encouraging invariance to the transformations within $\mathcal{B}_2$. We provide some empirical evidence for this in Table 5. When we recast the module to minimise $L_2\big(f(x_1), f(x'_1)\big)$ ($L_2$ being the mean squared error loss), performance decreases notably on all datasets, with an average relative decrease of over 8%. This is because, with this loss, we have enforced invariance to the transformations in $\mathcal{B}_2$. In particular, we have encouraged invariance to all the elements of an affine transformation, which proves problematic.

Table 5: Performance for SimCLR for transformation invariant and non-transformation invariant representations using the affine objective.

|  | CIFAR10 | CIFAR100 | SVHN | '6' vs '9' |
|---|---|---|---|---|
| Invariant | $61.43 \pm 0.0053$ | $26.13 \pm 0.0074$ | $80.32 \pm 0.0047$ | $68.64 \pm 0.0085$ |
| Not Invariant | $\mathbf{64.71 \pm 0.0023}$ | $\mathbf{31.33 \pm 0.0024}$ | $\mathbf{83.85 \pm 0.0017}$ | $\mathbf{72.35 \pm 0.0058}$ |

To delve deeper into the effect of transformation invariance on performance, we extract only the '6' and '9' classes of the SVHN dataset as a new dataset and repeat the SSL pre-training and linear evaluation tasks. The goal of this experiment is to observe how performance degrades when the neural network is encouraged to be invariant to certain transformations - including rotation - in a setting where certain invariance (i.e. rotation) is not desirable. Results can also be seen in Table 5. This further suggests that invariance to certain transformations is not always desirable. Evidence from Table 5 suggests that transformation invariance (for this particular class of transformations) in SSL may not always be desirable, and may, in fact, hurt performance, even when this may not be expected (as in the case with CIFAR10 and CIFAR100, since no classes of these seem as if they should be affected by transformation invariance like the '6' vs '9' case). For more details on the invariance analyses, see Appendix C.

## 4.4 TRANSFORM COMPONENT ANALYSIS

Table 6 shows the performance of the various components of an affine transformation in terms of linear evaluation accuracy on the dataset.[2] To compute these results, the output dimensionality of mapping $h$ needs to be changed accordingly. Namely, rotation, translation, scale, and shear have corresponding output dimensionalities $m$ of 1, 2, 1, and 2, respectively. Interestingly, shear alone outperforms the three other transforms on all datasets for both SimCLR and BYOL. We hypothesise that this is because shear corrupts the image the most out of the four transforms, but still in a recognisable way. This forces the networks to learn more complex geometry and information about the object that the other transforms. We leave further investigation of this to future work.

Table 6: Performance comparison of the components of an affine transformation for SimCLR and BYOL. Best-performing transformation highlighted in bold for each dataset.

|  | SimCLR | | | BYOL | | |
|---|---|---|---|---|---|---|
|  | CIFAR10 | CIFAR100 | SVHN | CIFAR10 | CIFAR100 | SVHN |
| Rotation | $63.57 \pm 0.0031$ | $29.40 \pm 0.0035$ | $83.46 \pm 0.0036$ | $58.66 \pm 0.0020$ | $26.46 \pm 0.0042$ | $76.34 \pm 0.0073$ |
| Translation | $64.36 \pm 0.0036$ | $29.83 \pm 0.0008$ | $82.60 \pm 0.0053$ | $58.45 \pm 0.0038$ | $25.74 \pm 0.0022$ | $77.50 \pm 0.0059$ |
| Scale | $64.05 \pm 0.0035$ | $30.34 \pm 0.0027$ | $82.80 \pm 0.0037$ | $59.12 \pm 0.0034$ | $25.47 \pm 0.0023$ | $76.53 \pm 0.0102$ |
| Shear | $\mathbf{64.56 \pm 0.0051}$ | $\mathbf{31.05 \pm 0.0033}$ | $\mathbf{84.25 \pm 0.0020}$ | $\mathbf{60.21 \pm 0.0048}$ | $\mathbf{26.94 \pm 0.0038}$ | $\mathbf{77.69 \pm 0.0059}$ |

## 4.5 ADDITIONAL ABLATIONS

We perform various additional experiments to motivate the choice of architecture. We experiment with other means of encoding the latent transformation, specifically, concatenation instead of vector difference. However, this results in marginal performance gains of an average of 0.28 percentage points across the 3 datasets. These results do not seem to justify the noticeable additional computational cost from the transformation representation being twice the size. For primarily this reason,

---

[2]Due to apparent instability in training BYOL with our module for these low-dimensional outputs (e.g. single real value output for rotation and scale), we temporarily replace MSE with logcosh, which stabilises training in this setting

we opt to stick with vector difference. Further, we experiment with having the module operate on the output of $g$ instead of $f$. Performance degrades for all datasets (SimCLR): 64.38 for CIFAR10, 29.99 for CIFAR100, and 82.49 for SVHN. Lastly, we perform some preliminary experiments into having two modules: one operating on $x_1$ and the other operating on $x_2$ (instead of just one module as per our original experimental setup). The resulting performance difference is negligible with this setup: CIFAR10 $65.28 \pm 0.61$, CIFAR100 $31.68 \pm 1.04$, and SVHN $84.10 \pm 0.23$. We posit that this is because if the one module can solve the homography estimation for $x_1$, then a module operating on $x_2$ will have to be able to solve the homography estimation for it, since the same types of random homographies/affine transformations are being applied to both streams.

## 5 CONCLUSION

Network size and time of training is a bottleneck in modern SS architectures that can compete on a performance level like supervised alternatives. We have shown that the proposed module that regresses the parameters of an affine transformation or homography as an additional objective assists this training bottleneck with faster convergence and better performance. The architecture of the module does not encourage invariance to the affine or homographic transformation, as invariance has been previously shown to be potentially harmful (Lee et al., 2020). Rather, the proposed module encourages these transformations to be encoded within the latent space itself by directly estimating the parameters of the transformation. Lastly, we note that the affine transformation performs better in all cases than the full homography, even though the homography is a superset of affine transformations. The experiments suggest that the additional ability of perspective transform in a homography does not yield any tangible benefit over a regular affine transformation in such low-resolution settings.

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

## A    DATA AUGMENTATION DETAILS

Tables 7 and 8 detail the parameter values and value ranges used in the experiments for the base transformation sets $\mathcal{B}_1$ and $\mathcal{B}_2$, respectively. The transformations from $\mathcal{B}_1$ are applied with a specified probability. We also normalise the parameters of the affine transformations in the following way. Consider a rotation angle $\alpha$, translation values $t_x, t_y$, and shear angles $s_x, s_y$. We perform the following normalisation on these parameters: $\alpha := \alpha/360$; $t_x := t_x/W$; $t_y := t_y/H$; $s_x := s_x/s_{\max}$; $s_y := s_y/s_{\max}$, where $H, W$ are the image height and width, respectively, and $s_{\max}$ is the maximum allowed shear.

## B    AFFINE VS HOMOGRAPHY

In addition to the perspective distortion factor of 0.5, we perform a sweep across this parameter for the values $\{0.1, 0.2, 0.8\}$. The results can be seen in Table 9. Interestingly, most distortion factors perform similarly on these datasets, with a distortion factor of $0.5$ performing best on average. However, when the factor gets too large, as is the case for $0.8$, then the images become too corrupted for the neural network to seemingly learn anything useful.

Table 7: Parameter values for base transformation set $\mathcal{B}_1$.

| Transformation | Parameter Value | Probability |
|---|---|---|
| Colour Jitter (brightness) | 0.8 | 0.8 |
| Colour Jitter (contrast) | 0.8 | 0.8 |
| Colour Jitter (saturation) | 0.8 | 0.8 |
| Colour Jitter (hue) | 0.2 | 0.8 |
| Random Resized Crop | (0.08, 1.0) | 1 |
| Random Horizontal Flip | - | 0.5 |
| Random Grayscale | - | 0.2 |
| Gaussian Blurring (kernel size) | $3 \times 3$ | 1 |
| Gaussian Blurring (variance) | $[0.1, 2]$ | 1 |

Table 8: Parameter values for base transformation set $\mathcal{B}_2$.

| Base Transformation | Parameter Value |
|---|---|
| Rotation | $[-90, 90]$ |
| Translation ($x$ and $y$) | $[0\%, 25\%]$ |
| Scaling | $[0.7, 1.3]$ |
| Shear ($x$ and $y$) | $[-25, 25]$ |
| Perspective | 0.5 |

Table 9: Sweep across the distortion factor for the homography using SimCLR.

| | CIFAR10 | CIFAR100 | SVHN |
|---|---|---|---|
| Factor = 0.8 | $57.17 \pm 0.0319$ | $23.13 \pm 0.0089$ | 78.67 |
| Factor = 0.5 | $\mathbf{64.04 \pm 0.0029}$ | $29.1 \pm 0.0025$ | $82.38 \pm 0.0024$ |
| Factor = 0.2 | $62.82 \pm 0.0094$ | $\mathbf{29.39 \pm 0.0028}$ | $82.73 \pm 0.0030$ |
| Factor = 0.1 | $62.45 \pm 0.152$ | $29.07 \pm 0.0035$ | $\mathbf{82.89 \pm 0.0014}$ |

## C    INVARIANCE ANALYSIS

Figures 4 and 5 shows the confusion matrices for a particular run of the SVHN dataset for SimCLR when enforcing (and not enforcing) transformation invariance. Interestingly, transformation invariance negatively affects most classes of the dataset. Unsurprisingly, the classes '6' and '9' are most negatively affected when transformation invariance is enforced. Rotation invariance in this context is prohibitive, and performance subsequently drops. By recasting the module as proposed - encode the transformation and predict its parameters - we do not enforce invariance, and instead allow the network to learn from a richer supervision signal by learning to estimate a homography.

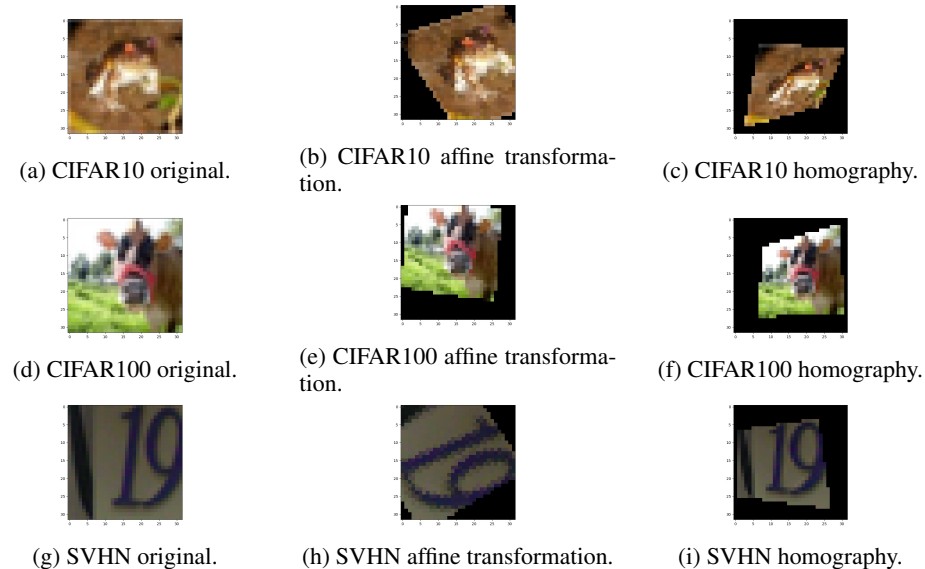

(a) CIFAR10 original.

(b) CIFAR10 affine transformation.

(c) CIFAR10 homography.

(d) CIFAR100 original.

(e) CIFAR100 affine transformation.

(f) CIFAR100 homography.

(g) SVHN original.

(h) SVHN affine transformation.

(i) SVHN homography.

Figure 3: Example affine transformations and homographies for all considered datasets.

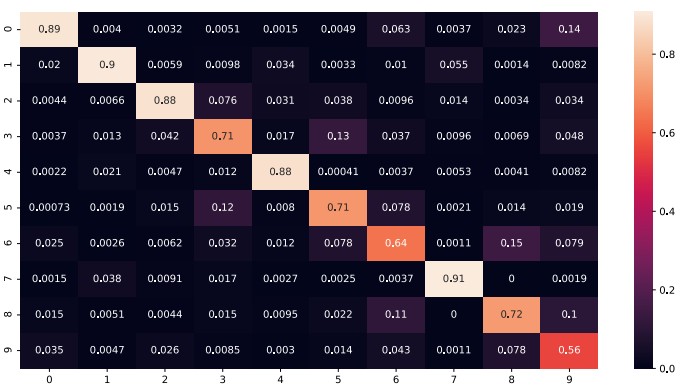

Figure 4: Confusion matrix on SVHN when enforcing transformation invariance.

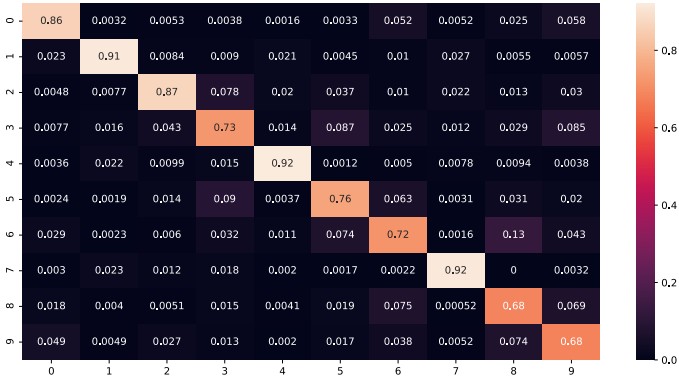

Figure 5: Confusion matrix on SVHN when not enforcing transformation invariance.

