# OpenReview forum: "Explicit homography estimation improves contrastive self-supervised learning"
_ICLR.cc/2021/Conference — Reject_

### Official Review · AnonReviewer1 · 2020-10-26
**Official Blind Review #1**

**Rating:** 4
**Confidence:** 5

**Review:**

**1. Summary:**

The authors propose a module that regresses the parameters of an affine transformation or homography as an additional objective in the contrastive self-supervised learning framework. The authors argue that the geometric information encoded in the proposed module can supplement the signal provided by a contrastive loss, improving both performance and convergence speed. The authors validate their claims with two recent contrastive self-supervised learning approaches (i.e., SimCLR, BYOL) on several benchmark datasets showing effective results.

**2. Strengths:**

+ The authors extend the typical contrastive objective with an additional homography estimation objective to enforce non-invariance to the affine transformation or homography. This extension is well motivated, and the authors provide a clear intuition for the proposed module.

+ The authors evaluate the proposed module on several benchmarks showing nice results.

**3. Weaknesses:**

While the authors demonstrate the effectiveness of the proposed module on several considered datasets, some points make me concerned about the actual usefulness of this module:

- The authors conduct the empirical study on three small-scale datasets (i.e., CIFAR10, CIFAR100, SVHN). This is fine in itself, however, without experiments on large-scale datasets like ImageNet, it is unknown whether the conclusion still holds since most of the previous effective contrastive learning methods (e.g., SimCLR, BYOL) are originally conducted on ImageNet. It would be better for the authors to provide such experiments to further validate the claims. In this case, there is no need to perform experiments with large batch size to achieve state-of-the-art performance, just a normal batch size (e.g., 256) can validate the effectiveness of the proposed module.

- The authors show significant improvements across all datasets with the proposed module in Table 2 and Table 3. However, as shown in Table 4, when pretrained for longer epoches, the relative benefit diminishes and even degrades the performance a little. Although the authors give a brief explanation on this phenomenon, I am concerned whether the proposed module is actually useful when SimCLR and BYOL have sufficiently converged.

- The authors use the linear evaluation to measure the quality of learned representations by pre-training and fine-tuning on the same dataset. Although it is one of the common evaluation protocols, pre-training and fine-tuning on the same dataset may induce some inevitable biases. Therefore, it would be better to provide more empirical evidence by transferring to other datasets to evaluate the effectiveness of the proposed module.

To sum up, although the paper is interesting, I think it requires more work (see points W1-W3 above) in order to become more complete and convincing. Therefore, I am leaning towards rejection.

====================================================================================

**Post-Rebuttal**

After reading the rebuttal and the other reviewers' comments, my concerns persist:

- The technical contribution is limited. Adding a pretext task of homography prediction itself brings little insights regarding how it improves upon contrastive representation learning from a different perspective.

- The experimental results are not convincing compared to the recent advances. The authors are encouraged to include ImageNet results as well as transfer learning evaluation.

Therefore, I would like to keep my initial rating as rejection.

---

> ### Author Response · Authors · 2020-11-13
> **Reviewer 1: Rebuttal**
>
> We thank the reviewer for the time taken in reviewing our work, and useful suggestions for improvements of the paper.
>
> Q1: The authors conduct the empirical study on three small-scale datasets (i.e., CIFAR10, CIFAR100, SVHN). This is fine in itself, however, without experiments on large-scale datasets like ImageNet, it is unknown whether the conclusion still holds since most of the previous effective contrastive learning methods (e.g., SimCLR, BYOL) are originally conducted on ImageNet. It would be better for the authors to provide such experiments to further validate the claims. In this case, there is no need to perform experiments with large batch size to achieve state-of-the-art performance, just a normal batch size (e.g., 256) can validate the effectiveness of the proposed module.
>
> A1: We agree with the reviewer that ImageNet is an ideal experiment to include in the paper. However, we were unable to include it due to computational resource limitations. We will be running these experiments in future work.
>
>
> Q2: The authors show significant improvements across all datasets with the proposed module in Table 2 and Table 3. However, as shown in Table 4, when pretrained for longer epoches, the relative benefit diminishes and even degrades the performance a little. Although the authors give a brief explanation on this phenomenon, I am concerned whether the proposed module is actually useful when SimCLR and BYOL have sufficiently converged.
>
> A2: We understand the reviewer's concern. However, we argue that at 500 epochs (see Table 4), the methods have indeed converged for these small-scale datasets. This is evidenced by the fact that for a much more difficult dataset (ImageNet), SimCLR was only trained for 100 epochs.
>
> Further, a primary aim of our work is to demonstrate the fact that being able to predict a homography/affine transformation as an additional objective provides the network with additional information during learning not available to it from the contrastive loss (particularly early on in learning). We opine that this is a useful finding, and merits further investigation into ways to improve or supplement contrastive SSL methods.
>
>
> Q3: The authors use the linear evaluation to measure the quality of learned representations by pre-training and fine-tuning on the same dataset. Although it is one of the common evaluation protocols, pre-training and fine-tuning on the same dataset may induce some inevitable biases. Therefore, it would be better to provide more empirical evidence by transferring to other datasets to evaluate the effectiveness of the proposed module.
>
> A3: We definitely agree with the reviewer. As per response A1, we will in future work be conducting experiments on ImageNet, including transfer learning experiments to be consistent with previous work. We were not able to perform these due to computational resource limitations.

---

### Official Review · AnonReviewer2 · 2020-10-26
**Adding a secondary loss on contrastive SSL that explicitly predicts an affine augmentation**

**Rating:** 4
**Confidence:** 3

**Review:**

====================================================

**Update after rebuttal**

I want to thank the authors for a long and highly detailed rebuttal. They clarified a lot of my questions and hopefully in the process they were able to improve the paper. However, my listed weaknesses still stand:
* there is no transfer learning experiments
* there not seem to be any consistent gains with the proposed approach over other methods, as seen in Table 4, after 500 epochs (when the models indeed have probably converged). Gains can be seen for 100 epochs, but from the absolute numbers  it is obvious that models havent really convered at that time. Furthermore, as the authors clarified,  they do require  30% higher training time, something that make the claim that they learn faster even weaker
* There is no analysis It is unclear why/how forcing the feature vectors difference to be predictive (ie encode) the transformation and learning representations that are non-invariant to homography transforms actually helps for learning better representations. In fact, It turns out that simpler versions of the proposed module than homography estimation are the ones giving the best results.

I will therefore retain my rejection rating.


====================================================

Summary:

The authors propose a module and objective that can be added to recent contrastive self-supervised learning (SSL) frameworks like SimCLR or BYOL. The module p`erforms an affine or homography transformation $\phi$ on the input $x_1$. Then the difference of the features of $x_1$ and the geometrically transformed version are the input to an MLP that tries to regress to the parameters of  transformation $\phi$, using an MSE loss. The authors show gains on CIFAR and SVHN by adding this module over SimCLR and BYOL.

Strong points: The authors show that adding this module speeds up SSL pre-training during early epochs.


Weak points:

A) The authors do not measure transfer learning performance and evaluate *only* on a very superficial setup, where they train linear classifiers on top of encoders that are pre-trained with contrastive SSL learning on the exact same datasets.  Although this is indeed an experiment popular among SSL papers,  in all recent SSL works like SimCLR and BYOL the linear evaluation on the same dataset (usually imagenet) is only a small part of the evaluation suite that also involves testing performance on many different tasks and datasets. Transfer learning performance here is not measured, nor is performance on  semi-supervised learning tasks.

B) The authors show some gains and faster learning for the first epochs of SSL pre-training. However, results should be taken with a pinch of salt as their gains are only visible in a realm where performance is still clearly suboptimal and learning is far from converged (eg for CIFAR100,  31.33 at epoch 100 vs 42.53 at epoch 500) To their credit, the authors themselves clearly say this in the text and show in Table 4. However they only present results at 100 and then 500 epochs, so it is hard to understand when (at which epoch) the two lines "cross" - at which epoch does the proposed approach stop giving gains?

C) There is no analysis It is unclear why/how forcing the feature vectors difference to be predictive (ie encode) the transformation and learning representations that are non-invariant to homography transforms actually helps for learning better representations. In fact, It turns out that simpler versions of the proposed module than homography estimation are the ones giving the best results. In Table 5, the authors show that most of the gain can be achieved by only predicting the 2-dim sheer tranformation; this is something potentially interesting that the authors however do not investigate any further.

Questions and notes:
* Although not used for SSL, the authors should cite and discuss the Spatial Transformer module of [Jaderberg et al 2015]
* The notation could be clearer (eg although clearly explained in the text, it is a bit confusing that augmentation/transformation $a_2$ is sampled from set $T_1$ and $a_\phi$ from $T_2$)
* Design ablations: The authors use vector difference as the input to MLP h to regress to the transformation parameters, but other reasoning modules could also be used. How would performance be affected for other input functions beyond difference, eg concatenation or elementwise multiplication?
* There area number of hyperparameter choices in Table 1 that do not match the corresponding papers (eg BYOL uses LARS not SGD). Why wasn't the original protocol followed, and were all the differences ablated?
* How do you balance the two losses in Eq(5)? Is there no hyperparameter? what would happen if you favor one or the other loss?
* How would performance change if this module was added after g() instead of f()?
* Why only on $x_1$? Could there be extra gains if the MSE loss was also applied over $x_2$?
* What is the added cost of the module in terms of training time?
* It is unclear to me what the variance measures: Is it for results after multiple runs, and if so, how many runs? Were the multiple runs from scratch or only for the linear evaluation?
* In Figure 2, how often was performance tested? markers would help understand where datapoints are.
* The datasets used are relatively small and more importantly in very low resolution. Would the same hold for higher resolution datasets?
* How dataset biased are the results presented? E.g. invariance to homographies might not be beneficial to digits (6 and 9 is a good example) but it would be useful in eg landmark datasets. Maybe a more suiting dataset/task would be something related to localization
* [Tian et al 2020] is another related paper that could be discussed - there the authors explore different augmentations to get invariance to, and show highly increased performance even after longer training.

References:
[Jaderberg et al 2015] Spatial transformer networks." Advances in neural information processing systems. 2015.
[Tian et al 2020] What makes for good views for contrastive learning, Arxiv May 2020, accepted at NeurIPS 2020 (not officially published yet)

---

> ### Author Response · Authors · 2020-11-13
> **Reviewer 2: Rebuttal**
>
> We thank the reviewer for the time taken in reviewing our work, and useful suggestions for improvements of the paper.
>
> Q1: The authors do not measure transfer learning performance and evaluate only on a very superficial setup, where they train linear classifiers on top of encoders that are pre-trained with contrastive SSL learning on the exact same datasets. Although this is indeed an experiment popular among SSL papers, in all recent SSL works like SimCLR and BYOL the linear evaluation on the same dataset (usually imagenet) is only a small part of the evaluation suite that also involves testing performance on many different tasks and datasets. Transfer learning performance here is not measured, nor is performance on semi-supervised learning tasks.
>
> A1: We agree with the reviewer that ImageNet is an ideal experiment to include in the paper. However, we were unable to include it due to computational resource limitations. We will be running these experiments in future work, including transfer learning experiments to be consistent with previous work.
>
>
> Q2: The authors show some gains and faster learning for the first epochs of SSL pre-training. However, results should be taken with a pinch of salt as their gains are only visible in a realm where performance is still clearly suboptimal and learning is far from converged (eg for CIFAR100, 31.33 at epoch 100 vs 42.53 at epoch 500) To their credit, the authors themselves clearly say this in the text and show in Table 4. However they only present results at 100 and then 500 epochs, so it is hard to understand when (at which epoch) the two lines "cross" - at which epoch does the proposed approach stop giving gains?
>
> A2: We understand the reviewer's concern. However, we argue that at 500 epochs, datasets such as CIFAR10, CIFAR100, and SVHN have indeed converged. Previous works (e.g. SimCLR) train for 100 epochs on the much more complicated ImageNet dataset. We will run experiments to train the methods for longer (1000 epochs) to verify this hypothesis. We will report results once they're in.
>
>
> Q3: There is no analysis It is unclear why/how forcing the feature vectors difference to be predictive (ie encode) the transformation and learning representations that are non-invariant to homography transforms actually helps for learning better representations. In fact, It turns out that simpler versions of the proposed module than homography estimation are the ones giving the best results. In Table 5, the authors show that most of the gain can be achieved by only predicting the 2-dim sheer tranformation; this is something potentially interesting that the authors however do not investigate any further.
>
> A3: We definitely agree with the reviewer. We did indeed previously consider concatenation as an alternative way to represent the transformation instead of vector difference. However, from preliminary experiments the performance difference was negligible. We are currently scaling the concatenation experiments to get final numbers on this.
>
> Regarding the reviewer's concern about the shear transformation outperforming SimCLR on SVHN, we would like to note that this marginal performance improvement of shear only occurred in 1 of 6 cases. Further, from visual inspection of samples from the SVHN dataset, we posit that shear appears to the most useful transformation to be able to encode for such house number images. We plan to investigate the transformation component analysis in future work.

---

> > ### Comment · AnonReviewer2 · 2020-11-20
> > **One clarification**
> >
> > Thank you for a detailed rebuttal.
> >
> > One quick clarification for your A2:
> >
> > "at 500 epochs, datasets such as CIFAR10, CIFAR100, and SVHN have indeed converged" - I agree with this statement, however, after 500 epochs (when the models indeed have probably converged) there not seem to be any consistent gains with your approach over other methods, as seen in Table 4. And that was my concern here. Let me know if I misunderstood something.
> >
> > "Previous works (e.g. SimCLR) train for 100 epochs on the much more complicated ImageNet dataset."
> > Although SimCLR has ablations at 100 epochs:
> > a) the also have results that verify their results on large scale and longer trainings as well, they don't just report ablations at 100 epochs.
> > b) A minor comment: please note that from Fig 9 in [SimCLR], training for 100 epochs (on Imagenet) with a batch size of 256 (which is the case in your experiments) results are highly suboptimal, and present results for up to 4k batch-size for 100 epoch training, something that makes a big difference.

---

> > > ### Author Response · Authors · 2020-11-20
> > > **Clarification of research aims**
> > >
> > > Thank you for your response. Hopefully I can provide some clarification below:
> > >
> > > Indeed at 500 epochs on these dataset there are no consistent gains, however, we aim to show primarily that: using the propose homography estimation objectives allows for faster learning. It allows for convergence to a particular linear evaluation accuracy in a shorter amount of epochs than with only the contrastive objective. The homography estimation provides a signal for supervision that contains additional information that is not available from the contrastive objectives early in learning. We feel this is useful to know, as it speaks to the efficiency of current contrastive SSL methods.
> > >
> > > Indeed they did present results with additional ablation to the 100 epoch setting. However, we posit that ImageNet pretraining requires these high batch sizes and additional epochs. For the datasets we have covered, higher batch sizes did not yield performance gains in preliminary experiments thereto. The batch size used in our experiments seemed to be fairly optimal, and number of epochs had a more significant effect on downstream performance with these datasets.
> > >
> > > We are currently running experiments on ImageNet to test our module on a large-scale, higher-resolution benchmark. However, due to computational resource limitations, this will take some time.
> > >
> > > Thank you, and please let us know if you require any further clarification.

---

> ### Author Response · Authors · 2020-11-13
> **Reviewer 2: Questions and notes rebuttal (part 1)**
>
> We thank the reviewer for the time taken in reviewing our work, and useful suggestions for improvements of the paper.
>
> Q1: Although not used for SSL, the authors should cite and discuss the Spatial Transformer module of [Jaderberg et al 2015]
>
> A1: We see the link with spatial transformers as being the fact that we're also regressing the parameters of a transformation somewhere in the network. Beyond that, we don't see any clear link. Could the reviewer clarify the reason to include this work?
>
>
> Q2: The notation could be clearer (eg although clearly explained in the text, it is a bit confusing that augmentation/transformation $a_2$ is sampled from set $T_1$ and $a_{\phi}$ from $T_2$)
>
> A2: We thank the reviewer for pointing this out. We will change the text to make the notation clearer.
>
>
> Q3: Design ablations: The authors use vector difference as the input to MLP h to regress to the transformation parameters, but other reasoning modules could also be used. How would performance be affected for other input functions beyond difference, eg concatenation or elementwise multiplication?
>
> A3: We definitely agree with the reviewer. We did indeed previously consider concatenation as an alternative way to represent the transformation instead of vector difference. However, from preliminary experiments the performance difference was negligible. We are currently scaling the concatenation experiments to get final numbers on this.
>
>
> Q4: There area number of hyperparameter choices in Table 1 that do not match the corresponding papers (eg BYOL uses LARS not SGD). Why wasn't the original protocol followed, and were all the differences ablated?
>
> A4: We thank the reviewer for pointing this out. The only difference between the original protocols are the optimiser-related hyperparameters. In preliminary experiments, we did indeed trial LARS with original hyperparameter values, however performance was significantly worse (borderline usable). Adam performed much better with far less hyperparameter tuning on these small-scale benchmarks. We posit that this is due to the hyperparameters in the paper being optimised for ImageNet, whereas datasets such as CIFAR10, CIFAR100, and SVHN require significantly different hyperparameters to train. In ImageNet experiments that we will be doing in future work, we will be using the original protocols.
>
>
> Q5: How do you balance the two losses in Eq(5)? Is there no hyperparameter? what would happen if you favor one or the other loss?
>
> A5: We agree with the reviewer that this is indeed a useful study to perform. We plan to do this in future work involving multiple kinds of downstream tasks (object detection and semantic segmentation). We posit that because the SSL algorithms with and without the module eventually appear to converge, balancing of the losses may need to be based on a schedule that favours one term of the loss more than the other over time.
>
>
> Q6: How would performance change if this module was added after g() instead of f()?
>
> A6: We definitely agree with the reviewer. We did indeed previously consider investigate using $g$ instead of $f$ for this. However, our experiments showed that the performance difference using $g$ is worse: 64.38 on CIFAR10; 29.99 on CIFAR100; and 82.49 on SVHN using SimCLR. We will add these findings to the appendix.
>
>
> Q7: Why only on $x_1$? Could there be extra gains if the MSE loss was also applied over $x_2$?
>
> A7: We thank the reviewer for pointing this out. We did instead trial adding our module to both $x_1$ and $x_2$. However, the performance different was negligible. We posit that this is because if the one module can solve homography estimation for $x_1$, then a module operating on $x_2$ will have to be able to solve homography estimation for it, because the same types of random homographies/affine transforms are being applied to both.
>
>
> Q8: What is the added cost of the module in terms of training time?
>
> A8: Total SSL training time is 30% larger on average with our module. We will systemically profile the training to identify where the bottleneck lies, and update you accordingly.
>
>
> Q9: It is unclear to me what the variance measures: Is it for results after multiple runs, and if so, how many runs? Were the multiple runs from scratch or only for the linear evaluation?
>
> A9: We thank the reviewer for pointing this out. This measure represents 10 random trials. Each run is a full pipeline trained from scratch: SSL pretraining + linear evaluation. We will clarify this in the paper.
>
>
> Q10: In Figure 2, how often was performance tested? markers would help understand where datapoints are.
>
> A10: Performance is tested every 10 epochs. We will add tick marks on to the plot, and clarify this in the text.

---

> ### Author Response · Authors · 2020-11-13
> **Reviewer 2: Questions and notes rebuttal (part 2)**
>
>
> Q11: The datasets used are relatively small and more importantly in very low resolution. Would the same hold for higher resolution datasets?
>
> A11: We agree with the reviewer in this regard. We planned to include ImageNet as part of our experiments (due to its larger scale in resolution and sample complexity) However, we were unable to include it due to computational resource limitations. We will be running these experiments in future work, including transfer learning experiments to be consistent with previous work.
>
>
> Q12: How dataset biased are the results presented? E.g. invariance to homographies might not be beneficial to digits (6 and 9 is a good example) but it would be useful in eg landmark datasets. Maybe a more suiting dataset/task would be something related to localization
>
> A12: We definitely agree with the reviewer in this regard. In future work we will be investigating the effect of invariance on a host of downstream tasks (object detection and semantic segmentation). However, we believe it is interesting to note that invariance to homographies/affine transformations in contrastive SSL does not yield any benefit in this low-resolution, small-scale regime of CIFAR10, CIFAR100, and SVHN.
>
>
> Q13: [Tian et al 2020] is another related paper that could be discussed - there the authors explore different augmentations to get invariance to, and show highly increased performance even after longer training.
>
> A13: This paper focuses on view generation, and the method proposed in the work focuses on learning effective views using ground truth label information from the downstream task. This label information is what informs the algorithm of effective views and implicitly learns how much invariance to employ. This is very different to our method, as our is fully unsupervised and we are not attempting to learn invariance. Thus it is difficult to draw any conclusion in comparisons with out work. We will, however, add this paper to the related work section.

---

### Official Review · AnonReviewer3 · 2020-10-28
**The paper needs stronger and more extensive experimentation results to be in a publicable state.**

**Rating:** 4
**Confidence:** 4

**Review:**

Summary: the paper introduces a novel technique for self-supervised learning where additionally to the self-supervised loss, the model is forced to predict a parameter of the homography between two images given the difference of its embedding vectors. Authors evaluate the proposed method in BYOL and SimCLR.

Strengths:

- Self-supervised learning has shown a lot of promise these last years. I believe that this work explores an interesting addition to traditional self-supervised methods.

- The paper evaluates this addition in two of the state-of-the-art self-supervised models (BYOL and SimCLR).

- Authors provide very detailed explanation of the experimental set-up, which is very good for reproducibility.


Weaknesses:

- In my opinion, the paper needs to evaluate the methods in more large-scale benchmarks. Authors only use CIFAR and street view house numbers dataset, which are very specific benchmarks. However, the community is moving towards using larger and larger benchmarks and I think that the paper should follow this trend to be publicable. I think authors should at least apply their method to ImageNet training.

- The performance improvement shown does not seem enough to merit publication, given the benchmark used. In particular, larger differences are shown only when one method does worse in absolute terms (Table 3). I think the combination of these two factors (the numerical improvement and the benchmarks used) make the paper weak.

- I think it would be useful for the reader to see some examples of the homography to estimate in the main paper. I believe it helps the reader understand the task the model is attempting and how hard or difficult this is?

- Have the authors considered using using a concatenation of the two vectors x1, x1' instead of its difference to predict the homography? I am missing some experiments motivating the choice of architecture.


Conclusion: I think the research direction is interesting but the paper needs to evaluate in larger dataset to show results that merit publication. Furthermore, I think authors should motivate better their architectural choices.

---

> ### Author Response · Authors · 2020-11-13
> **Reviewer 3: Rebuttal**
>
> We thank the reviewer for the time taken in reviewing our work, and useful suggestions for improvements of the paper.
>
> Q1: In my opinion, the paper needs to evaluate the methods in more large-scale benchmarks. Authors only use CIFAR and street view house numbers dataset, which are very specific benchmarks. However, the community is moving towards using larger and larger benchmarks and I think that the paper should follow this trend to be publicable. I think authors should at least apply their method to ImageNet training.
>
> A1: We agree with the reviewer that ImageNet is an ideal experiment to include in the paper. However, we were unable to include it due to computational resource limitations. We will be running these experiments in future work, including transfer learning experiments to be consistent with previous work.
>
>
> Q2: The performance improvement shown does not seem enough to merit publication, given the benchmark used. In particular, larger differences are shown only when one method does worse in absolute terms (Table 3). I think the combination of these two factors (the numerical improvement and the benchmarks used) make the paper weak.
>
> A2: We understand the reviewer's concerns. However, we would like to note that the experiments were statistically significant for 10 random trials in all experiments with a significance level of 1%. Further, in our work we primarily aim to show that being able to predict a homography/affine transformation provides the network with supplementary information during learning that is not available from the contrastive loss, particularly early in learning. We feel that this is useful information to know about the current state of contrastive SSL methods, and may assist in directing future work.
>
>
> Q3: I think it would be useful for the reader to see some examples of the homography to estimate in the main paper. I believe it helps the reader understand the task the model is attempting and how hard or difficult this is?
>
> A3: We agree with the reviewer. We will include example homographies and affine transformations of varying strengths in the paper.
>
>
> Q4: Have the authors considered using using a concatenation of the two vectors x1, x1' instead of its difference to predict the homography? I am missing some experiments motivating the choice of architecture.
>
> A4: We definitely agree with the reviewer. We did indeed previously consider concatenation as an alternative way to represent the transformation instead of vector difference. However, from preliminary experiments the performance difference was negligible. We are currently scaling the concatenation experiments to get final numbers on this.

---

### Official Review · AnonReviewer4 · 2020-10-28
**The paper experiments with H and A estimation as an additional loss in the image recognition task. Marginal improvement is reported.**

**Rating:** 4
**Confidence:** 3

**Review:**

The paper test a hypothesis that adding  affine transformation and homography estimation as an auxilliary loss in an image recognition task will improve performance. In experiments on CIFAR10, CIFAR100 and SVHN, improvement of a up to a few % points is observed.

The technique is close to augmentation and experimenting with A and H is not novel.

There are issues with the experiments:
1. a certain representation of A and H was chosen. Other decompositions are possible. Is the ad hoc chosen one suitable? Have other options been considered?
2. the representation of A and H influences the sampling.
3. the three problems used for demonstrations are quite similar. It is unclear what type of vision problems would benefit from this technique.

Parts of the paper state well-known facts (description of H and A, there properties), some are irrelevant - for instance par 2. in the intro about transfer learning. More details should be provided about SimCLR and BYOL to make the submission more self-contained.

I see very limited benefit to the reader.

---

> ### Author Response · Authors · 2020-11-13
> **Reviewer 4: Rebuttal**
>
> We thank the reviewer for the time taken in reviewing our work, and useful suggestions for improvements of the paper.
>
> Q1: a certain representation of A and H was chosen. Other decompositions are possible. Is the ad hoc chosen one suitable? Have other options been considered?
>
> A1: We previously considered concatenation as an alternative way to represent the transformation instead of vector difference. However, from preliminary experiments the performance difference was negligible. We are currently scaling the concatenation experiments to get final numbers on this.
>
>
> Q2: the representation of A and H influences the sampling.
>
> A2: We assume the reviewer means that we are predicting the actual individual parameters of an affine transform, instead of the transformation matrix elements. We plan to run experiments to analyse how the latter compares that what we've already implemented in the former.
>
>
> Q3: the three problems used for demonstrations are quite similar. It is unclear what type of vision problems would benefit from this technique.
>
> A3: In future work we plan to investigate the benefit of the proposed module on the following downstream tasks: objection detection, and semantic segmentation.
>
>
> Q4: I see very limited benefit to the reader.
>
> A4: The main benefit we aim to show with the proposed module is that being predictive of a homography/affine transformation provides the network with supplementary information not available from the contrastive loss, particularly early in learning. We feel that this is useful to know, and to investigate in future work.

---

### Decision · Program_Chairs · 2021-01-07
**Final Decision**

**Decision:**

Reject

**Comment:**

All four reviewers raised concerns on the limited technical novelty and insufficient experiments. They unanimously recommended a rejection. I carefully read the authors' rebuttal but did not find strong reasons to go against the reviewers' recommendations. The reviewers made excellent points to further improve the paper. The authors are encouraged to incorporate those for a future submission.